# Pulmonary Toxicity and Proteomic Analysis in Bronchoalveolar Lavage Fluids and Lungs of Rats Exposed to Copper Oxide Nanoparticles

**DOI:** 10.3390/ijms232113265

**Published:** 2022-10-31

**Authors:** Jung-Taek Kwon, Yoonjin Kim, Seonyoung Choi, Byung-ll Yoon, Hyun-Sook Kim, Ilseob Shim, Donggeun Sul

**Affiliations:** 1Environmental Health Research Department, National Institute of Environmental Research, Incheon 22689, Korea; 2Graduate School of Medicine, Korea University, 73 Inchon-ro, Sungbuk-ku, Seoul 136-705, Korea; 3College of Veterinary Medicine and Institute of Veterinary Science, Kangwon National University, Chuncheon 24341, Korea; 4Department of Biomedical Laboratory Science, College of Health Science, Cheongju University, Cheongju 28503, Korea

**Keywords:** bronchoalveolar lavage fluid, copper oxide, nanoparticles, proteomics, rat lung

## Abstract

Copper oxide nanoparticles (CuO NPs) were intratracheally instilled into lungs at concentrations of 0, 0.15, and 1.5 mg/kg bodyweight to 7-week-old Sprague–Dawley rats. The cytotoxicity, immunotoxicity, and oxidative stress were evaluated, followed by proteomic analysis of bronchoalveolar lavage fluid (BALF) and lungs of rats. The CuO NPs-exposed groups revealed dose-dependent increases in total cells, polymorphonuclear leukocytes, lactate dyhydrogenase, and total protein levels in BALF. Inflammatory cytokines, including macrophage inflammatory protein-2 and tumor necrosis factor-α, were increased in the CuO NPs-treated groups. The expression levels of catalase, glutathione peroxidase-1, and peroxiredoxin-2 were downregulated, whereas that of superoxide dismutase-2 was upregulated in the CuO NPs-exposed groups. Five heat shock proteins were downregulated in rats exposed to high concentrations of CuO NPs. In proteomic analysis, 17 proteins were upregulated or downregulated, and 6 proteins were validated via Western blot analysis. Significant upregulation of 3-hydroxy-3-methylglutaryl-CoA synthase and fidgetin-like 1 and downregulation of annexin II, HSP 47 and proteasome α1 occurred in the CuO NPs exposed groups. Taken together, this study provides additional insight into pulmonary cytotoxicity and immunotoxicity as well as oxidative stress in rats exposed to CuO NPs. Proteomic analysis revealed potential toxicological biomarkers of CuO NPs, which also reveals the toxicity mechanisms of CuO NPs.

## 1. Introduction

Nanoparticles are materials measuring 1–100 nm in diameter with distinct physiochemical, biological and electrical properties due to their small size and large surface-to-volume ratio [1,2,3,4,5]. Due to the rapid growth of nanotechnology, nanoparticles have been used in industrial catalytic processes, energy conversion and storage, image display, cosmetics, medical theranostics, including analytical nano-devices, drug delivery and targeting nano-carriers, tissue engineering, and clinical and toxicological applications [1,2,3,4,5]. Nanoparticles can be divided into different groups based on their dimensionality, morphology, porosity, composition, origin, phase, and dispersion state [6]. Based on their chemical compositions, nanoparticles can be classified as carbon, inorganic, organic, or hybrid nanoparticles. Inorganic nanomaterials include metal oxide nanoparticles, metallic nanoparticles, and quantum dots [6,7]. Metal and metallic oxide nanoparticles are heterogeneous in nature. Their impact on living cells and tissues is attributed to their size and shape rather than the chemical behavior of specific metal ions used [7]. Metal and metal oxide nanoparticles such as gold, silver, zinc, nickel, platinum, zinc oxide, titanium dioxide, copper oxide (CuO), iron oxide, nickel oxide, and magnesium oxide nanoparticles were widely utilized in biomedical applications [1,2,3,4,5]. Owing to the superior properties and functional versatility, the metal and metal oxide nanomaterials have attracted a lot of research interest over the past few years [4]. 

Among them, CuO NPs have emerged as an important class of nanomaterials with a wide range of medicinal, environmental, and industrial applications, which are associated with a potential environmental risk and health hazard [8]. When compared with TiO_2_, ZnO, CuZnFe_2_O_4_, Fe_3_O_4_, and Fe_2_O_3_ NP, the cytotoxicity of CuO NPs was relatively higher and resulted in cell death and DNA damage [9,10].

In general, human exposure to engineered nanoparticles occurs via inhalation, dermal penetration, ocular exposure, and ingestion [5,11,12,13]. Previous studies reported the delivery of metal nanoparticles and metal oxide nanoparticles to the lungs and gastrointestinal tract via the respiratory and digestive tract [14]. Especially, the lungs were the first line of contact for particles entering the body and hence the most likely organ for accumulation and long-term exposure [15]. NP inhalation and pulmonary exposure are most common in humans, and lungs are one of the most vulnerable organs to damage [16,17,18].

Therefore, the cytotoxic, genotoxic, and immunotoxic effects of CuO NPs on lungs have been studied using human alveolar basal epithelial cells (A549) and non-tumorigenic lung epithelial cells (BEAS-2B) [19,20,21,22,23,24] as well as bronchoalveolar lavage fluid (BALF) and lung tissues of rats and mice [25,26,27,28,29,30,31,32]. Furthermore, genomic, proteomic, and metabolomic studies, including toxicokinetics, toxicodynamics, and toxicological pathways, were performed to elucidate the toxicity mechanisms of CuO NPs [9,19,21,22,26,33,34,35,36]. Specifically, toxico-proteomic analyses were conducted to analyze the differential gene expression at the protein level and identify critical proteins by comparing the proteomic patterns under different conditions after exposure to CuO NPs in human lung cells [9], mussel *Mytilus galloprovincialis* [35], and *Cucumis sativus* seeds [36]. Nevertheless, few proteomic studies involved pulmonary organs of animals exposed to CuO NPs [9,35,36]. 

In the present study, we used two doses of CuO NPs, including 0.15 and 1.5 mg/kg bodyweight (bw), under low and high exposure conditions in rats. In a previous intratracheal instillation study, CuO NPs measuring 20 nm in size induced significant inflammation at doses of 0.17 to 1.7 mg/kg bw instilled intratracheally into rats [28]. In addition, CuO NPs smaller than 50 nm induced significant cytotoxic effects in BALF and oxidative stress in lung tissues following intratracheal delivery of CuO NPs at doses of 1 to 5 mg/kg bw. Similar pulmonary toxicities were found under two exposure conditions a day after the instillation exposure [30]. 

For proteomic study, we measured the size of CuO NPs and then evaluated the cytotoxic and immunotoxic effects on BALF in terms of total cell count, the percentage of polymorphonuclear leukocytes (PMNs), total proteins, lactate dyhydrogenase (LDH), macrophage inflammatory protein-2 (MIP-2), and tumor necrosis factor (TNF)-α levels. Furthermore, histopathology and expression of antioxidant enzymes, including superoxide dismutase (SOD), catalase, glutathione peroxidase-1 (Gpx-1), peroxiredoxin-2 (Prx-2), and heat shock proteins (HSP), including HSP 10/40/70/90, were determined to evaluate the toxic effects of CuO NPs in lung tissues of rats exposed to CuO NPs. In the proteomic study, two-dimensional gel electrophoresis and mass spectrophotometry were used to compare the protein expression profiles of lung tissues in rats exposed to CuO NPs and validated via Western blot analysis.

Taken together, the present study provides additional insight into pulmonary cytotoxicity and immunotoxicity as well as oxidative stress in rats exposed to CuO NPs. Furthermore, the proteomic study revealed that the differentially expressed proteins could be used as potential toxicological biomarkers of CuO NPs and elucidated the toxicity mechanisms of CuO NPs.

## 2. Results

### 2.1. Characterizations of CuO NPs

In a previous study, CuO NPs (#544868, Lot #MKAA0633 from Sigma-Aldrich) were characterized with respect to chemical composition, crystallinity, morphology, endotoxin contamination, surface area, pH, and zeta potential (ZP) in cell culture media [37]. SEM showed an approximately spherical and smooth surface of CuO NPs, which exhibited a broad size distribution ranging from approximately 20 nm to around 200 nm, resulting in an average of 55 nm in diameter [37]. The present study showed similar results. TEM and SEM analyses revealed that CuO NPs had a primary size of around 20–80 nm and were partially aggregated into clusters (Figure 1).

### 2.2. Total Cell Count and the Percentages of PMNs, Lymphocytes, Eosinophils, and Macrophages in BALF of Rats Exposed to CuO NPs

Cellular profiles of BALF included alveolar macrophages, lymphocytes, neutrophils, and eosinophils. Neutrophils and eosinophils constitute PMNs in BALF. In general, lung edema, endothelial, and epithelial injuries are accompanied by an influx of neutrophils into the interstitial and broncheoalveolar space. Neutrophils are considered to play a key role in the progression of acute lung injury, which is characterized by activation and transmigration of neutrophils [38]. Total cell counts and the percentage of PMNs in BALF of rats exposed to CuO NPs were determined (Figure 2). Both total cell count and the percentage of PMNs in BALF increased significantly in exposed groups in a dose-dependent manner when compared with control groups. The increase in the percentage of PMNs in BALF was significant (Figure 2). The percentages of other inflammatory cells, including lymphocytes, eosinophils, and macrophages, in total cells in BALF were also determined (Figure 2). The percentage of macrophages was observed to decrease with the increasing concentration of CuO NPs. The percentage of eosinophils was not detectable in the control group, but detection in low and high exposed groups was increased to 1–3 %. There was no significant change in the percentage of lymphocytes between the control and exposed groups (Figure 2).

### 2.3. LDH Activity, Total Proteins and Inflammatory Cytokines in BALF of Rats Exposed to CuO NPs

The LDH activity and total proteins in BALF of rats exposed to CuO NPs were measured to evaluate the cytotoxicity and alveolar capillary membrane permeability [39], respectively. LDH activity and total proteins in BALF increased significantly in the exposed groups in a dose-dependent manner when compared with control groups. The increasing patterns of LDH activity and total proteins were similar in the exposed groups (Figure 3). The levels of inflammatory cytokines in BALF, including MIP-2 and TNF-α, were measured via ELISA in BALF of rats exposed to CuO NPs. The levels of MIP-2 and TNF-α increased significantly in BALF of rats exposed to high concentrations of CuO NPs (1.5 mg/kg bw) (Figure 3).

### 2.4. Histopathology of Lungs in Rats Exposed to CuO NPs

The lung tissues of the control group of rats did not show any histopathological changes. However, the CuO NPs-exposed group showed acute suppurative inflammation around the terminal bronchioles and alveoli. Specifically, pulmonary edema was detected in rats exposed to high concentrations of CuO NPs (1.5 mg/kg bw) but not in controls and rats exposed to low concentrations (0.15 mg/kg bw) (Figure 4).

### 2.5. Expression of Antioxidant Enzymes in Lung Tissues of Rats Exposed to CuO NPs

The levels of antioxidant enzymes, including SOD-2, catalase, Gpx-1, and Prx-2, were analyzed via Western blot to evaluate oxidative stress response. Significant downregulation of expression of catalase, Gpx-1, and Prx-2 occurred in lungs of rats exposed to high concentrations of CuO NPs (1.5 mg/kg bw) when compared with the control groups. However, the expression level of SOD-2 showed significant upregulation in response to increasing concentrations of CuO NPs (Figure 5).

### 2.6. Expression of HSP in Lung Tissues of Rats Exposed to CuO NPs

Based on their molecular weights, HSPs are classified into different families, such as HSP 40, HSP 60, HSP 70, HSP 90, HSP 110, and small HSPs [40]. In the present study, five HSPs, including HSP 10, HSP 40, HSP 70, HSC 70, and HSP 90, were conducted to evaluate the levels of oxidative stress in the lungs of rats exposed to CuO NPs. Exposure to a low concentration (0.15 mg/kg bw) did not alter the expression of any HSPs significantly when compared with the control groups. Specifically, HSC 70 and HSP 70 levels showed increased expression at low concentrations of CuO NPs. However, all HSPs were downregulated in lung tissues of rats exposed to high concentrations of CuO NPs (1.5 mg/kg bw) when compared with control groups (Figure 6).

### 2.7. 2-DE of Expressed Proteins in Lung Tissues Exposed to CuO NPs

Intratracheal instillation of CuO NPs at two different concentrations (0.15 mg/kg bw, 1.5 mg/kg bw) into SD rats was used to identify the toxicological marker proteins expressed in lung tissue. Two different pI strips (4–7 and 6–9) were used for protein analysis. The 2-DE patterns of proteins in lung tissues obtained using 4–7 and 6–9 pI strips showed 783 and 638 protein spots in the gels, respectively. A total of 1421 protein spots were resolved and then subjected to proteomic analysis (Appendix A). When compared with the controls in gels of 4–7 and 6–9 pI strips, 4 and 13 protein spots exhibited differential expression, respectively. Thus, a total of 17 protein spots (Figure 7 and Figure 8), including 7 upregulated and 10 downregulated proteins, were resolved in a dose-dependent manner.

### 2.8. Protein Identification and Validation

In the 1421 protein spots, 17 were upregulated or downregulated in lung tissues of rats exposed to CuO NPs. The protein spots identified via nano-ultra performance LC-ESI-MS^E^ are summarized in Table 1 (Appendix A). These proteins were involved in metabolism, transportation, protein regulation, oxidative stress response, cell signal transduction, immunity, cellular structure, and organization. Of the 17 proteins differentially expressed in lung tissue, a total of 6 proteins were validated via Western blot analysis. Significant upregulation of expression of cHMGCS and FIGNL 1 occurred in response to increasing concentrations of CuO NPs, whereas the expression of annexin II, HSP 47, and proteasome α1 was downregulated in response to increasing concentrations of CuO NPs (Figure 9).

## 3. Discussion

In the present study, we evaluated cytological and immunological changes in BALF and histopathological changes in the lungs of rats exposed to CuO NPs. Cellular profiles of BALF included alveolar macrophages, lymphocytes, neutrophils, and eosinophils. Specifically, neutrophils are considered to play a key role in the progression of acute lung injury, which is characterized by activation and transmigration of neutrophils [38]. A comparison of cellular profiles in the present and previous studies revealed similar results following CuO NPs exposure via intratracheal instillation in rats. An increase in the total number of cell count and a significant increase in primarily PMNs occurred with increasing CuO NPs exposure resulting in significant cytotoxicity [25,28,30]. In previous biochemical studies, the CuO NPs exposure did not result in similar percentages of control in both total proteins and LDH activity in BALF [28,30]. However, in the present study, the CuO NPs exposure led to a similar increase in both total proteins and LDH activity in BALF. Neutrophil infiltration of the lung is controlled by a complex network of chemokines that are released by a variety of cell types. In rodents, the most relevant chemokines for neutrophil recruitment into the lung are keratinocyte-derived chemokines or cytokine-induced neutrophil chemoattractant and macrophage inflammatory protein-2 (MIP-2) [38]. In the present study, CuO NPs induced the MIP-2 production, though the increasing rate of MIP-2 did not show significant fold changes compared with other cytotoxicity parameters in BALF of rats exposed to CuO NPs.

Acute lung injury is an uncontrolled pulmonary inflammation mediated by various inflammatory cells, such as PMNs and alveolar macrophages. Activated PMNs produce and secrete proinflammatory and anti-inflammatory cytokines to regulate inflammation. Excessive pro-inflammatory cytokines and insufficient anti-inflammatory cytokine levels trigger uncontrolled pulmonary inflammation [41]. In the present study, we determined the levels of two inflammatory cytokines, including TNF-α and MIP-2, to evaluate immunotoxicity in BALF of rats exposed to CuO NP. Exposed to low concentrations of CuO NP (0.15 mg /kg bw), TNF-α did not result in significant differences compared with the control groups. However, the level of MIP-2 showed a significant dose-dependent increase in the exposed groups. Specifically, the level of TNF-α revealed a significant increase in highly exposed groups when compared with control groups. A previous study, which showed the acute pulmonary immune toxicities of nanoparticles, also reported significant changes in IL-1β in BALF of rats exposed to CuO NPs (400 μg/rat) [42]. Our study has a limitation for elucidating inflammatory responses in rat lungs exposed to CuO NPs. Further studies are needed to investigate inflammatory responses by determining pro- and anti-inflammatory cytokines (IL-1β, IL-8, IL-4, IL-10, IL-13, and IFN-γ) and their balance.

Histopathological analysis revealed acute suppurative inflammation on day 1 around the terminal bronchioles and alveoli in a dose-dependent manner in rat lung tissues treated intratracheally with low and high doses of CuO NPs. In addition, pulmonary edema was diffusely observed in rat lung tissues at high doses of treatment. The present histopathological study was similar to previous reports [30,31]. The findings showed that intratracheal exposure to CuO NPs induced acute bronchioloalveolar inflammation with diffuse pulmonary edema, indicating pulmonary toxicity of CuO NPs. 

The expression levels of antioxidant enzymes (SOD2, catalase, Gpx-1, and Prx-2) and HSPs (HSP 10/40/70/90 and HSC 70) were evaluated to determine the oxidative stress in lung tissues of rats exposed to CuO NPs. The expression of catalase, Gpx-1, and Prx-2 was downregulated, whereas that of SOD2 was upregulated. Previous in vitro studies reported varying levels of antioxidant enzyme activities. In A549 cells exposed to CuO NPs for 24 h, SOD2 was upregulated, while most other genes associated with antioxidant defense mechanisms, including catalase and Gpx-2, were downregulated [22]. These results were similar to our present study. However, Prx-2 and 6 were upregulated in BEAS-2B cells exposed to CuO NPs for 24 h [9]. In the in vivo studies, CuO NPs-exposed rats showed a dose-dependent decrease in SOD and catalase levels in lungs more than in the control at all post-exposure periods [30]. However, the activities of catalase and SOD decreased significantly, whereas Gpx activity increased in erythrocytes of rats exposed to CuO NPs [43]. Furthermore, rats were treated with nano-CuO at a dose of 0.5 mg/(kg/day) via intraperitoneal injection over a period of 14 consecutive days. The activities of SOD and Gpx were statistically reduced in rat hippocampi exposed to CuO NPs [44]. In the cerebral cortex of rats exposed to 0.5 mg/kg bw of CuO NPs, a significant decrease in Gpx activity was found [45]. An in vitro proteomic study showed that HSPA9 and HSPE1 were significantly downregulated, whereas HSPA8 and HSPA1A were upregulated in BEAS-2B cells exposed to CuO NPs for 24 h. Peroxiredoxin-2 and 6 were upregulated by 2.1- and 2.6-fold, respectively [9]. In A549 cells, HSPA1A was almost equally enhanced by CuO NPs exposure at the transcriptional level to a maximum of 7-fold [46]. However, few studies were conducted to evaluate the oxidative stress involving HSPs in in vivo systems. In the present studies, the expression of five HSPs, including HSP 10, HSP 40, HSP 70, HSC 70, and HSP 90, was determined to evaluate the level of oxidative stress in the lungs of rats exposed to CuO NPs. Exposure to high concentrations resulted in downregulation of all HSPs, while HSC 70 and HSP 70 levels increased under low exposure. The reduction in HSP levels at high concentrations of CuO NPs may be related to threshold limits in the cell. Further studies are needed to elucidate the role of other HSPs mediated via antioxidant defense systems in organisms exposed to CuO NPs.

Proteomic analysis of 1421 proteins in the lungs of rats exposed to CuO NPs revealed significant upregulation and downregulation of 17 proteins. Of these, the expression of seven proteins, including Annexin II, HSP 47, 20s proteasome α1, HMGCS, and FIGNL, was validated via Western blot analysis.

The annexins are Ca^2+^-regulated, phospholipid- and membrane-binding proteins. AnxA2, which is a 36 kDa protein, is also known as p36, annexin II, ANXA2, calpactin I, lipocortin II, chromobindin VIII, or placental anticoagulant protein IV [47,48]. AnxA2 is produced by a wide spectrum of cell types, including endothelial, trophoblast, epithelial, and tumor cells, as well as innate immune cells, such as macrophages, monocytes, and dendritic cells [49]. This calcium-dependent binding and aggregation contributes to its biological functions, including vesicular transport, exocytosis, and endocytosis. Otherwise, ANXA2 participates in cell survival, proliferation, invasion and metastasis, thus acting as a regulator of tumor growth and progression, suggesting that ANXA2 is a target in cancer treatment [50]. In addition, ANXA2, as the substrate of thioredoxin, can also act as an endogenous antioxidant [51] and reduce inflammation via its role as an endogenous inhibitor of phospholipase A2 in mononuclear cells [52]. Specifically, recent studies show that secreted ANXA2 activates macrophages to induce the secretion of IL-1, IL-6, and TNF-α via TLR4/MyD88- and TRIF-dependent pathways [53]. The cytokines released by macrophages may promo-te tumor progression, such as IL-6 in pancreatic cancer and hepatoma [49]. In previous proteomics studies, ANXA2 was found to be upregulated in human BEAS-2B cells exposed to airborne nanoparticles with thermodynamic diameters less than 56 nm [54]. However, the expression of ANXA2 was significantly downregulated in human BEAS-2B cells exposed to titanium dioxide (12–88 nm) [55]. In the present study, ANXA2 was significantly downregulated in lung tissues of rats exposed to CuO NPs. CuO NPs may directly inhibit physiological, biochemical, and molecular regulatory functions via downregulation of ANXA2. However, the downregulation of ANXA2 by CuO NPs might prevent tumor progression and angiogenesis via inhibition of macrophage activation and secretion of IL-1, IL-6, and TNF-α. Further studies are required to investigate the pulmonary toxicity of CuO NPs in rats. 

HSP 47, which is encoded by the SERPINH1 gene, is a collagen-binding stress protein localized in the endoplasmic reticulum [56]. HSP 47 has been associated with various collagen-related diseases and may be a great target for fibrotic disorders [57]. In the present study, HSP 47 was significantly downregulated in lung tissues exposed to CuO NPs. A significant downregulation of other HSPs, including HSP 10/40/70/90, occurred in lung tissues of rats exposed to CuO NPs. Previous studies involving animal models of fibrosis, including pulmonary fibrosis, have shown that downregulation of HSP 47 expression by antisense oligodeoxynucleotides or by small interfering ribonucleic acid reduced collagen production and subsequently diminished progression of fibrosis [58]. The downregulation of HSP 47 by CuO NPs exposure might protect pulmonary cellular components against pulmonary fibrosis induced by CuO NPs.

Proteasome is a multi-catalytic proteinase catalyzing the degradation of the majority of intracellular proteins; it has a cylindrical structure comprising four stacked rings of α- and β-subunits [59]. In NP-treated cells, silica nanoparticles easily enter cells, translocate to the nucleus, and induce the formation of aberrant protein aggregates in the nucleus [60]. In microtubular networks of *Arabidopsis thaliana* exposed to TiO NP, tubulin misfolding was detected, followed by aggregate formation and increased proteasome-dependent degradation [61]. In the present study, CuO NPs significantly decreased the expression of 26S proteasome α1, which depleted 26S proteasome activity by inhibiting the degradation of protein aggregates induced by CuO NPs toxicity.

HMGCS, 3-hydroxy-3-methylglutaryl-CoA (HMG-CoA) synthase, catalyzes the condensation of acetoacetyl-CoA and acetyl-CoA to form HMG CoA plus free CoA. HMG-CoA synthase activity is located in two different compartments: the cytosol and the mitochondria. The HMG-CoA produced by the cytosolic HMG-CoA synthase (cHMGCS) is transformed into mevalonate by the action of HMG-CoA reductase [62,63]. In a previous proteomic study, albumin-bound nanoparticles, such as abraxane, was added to abraxane-sensitive (A549) and -resistant cells (A549/Abr) to investigate protein profiles. HMSCS was found to increase more than 2-fold in A549/Abr [64]. Additionally, the effect of nanoparticle-rich diesel exhaust on testosterone biosynthesis was evaluated in adult male mice. Subchronic, high-dose NR-DE exposure affected testosterone secretion and upregulated the mRNA expression of several important enzymes, such as HMG-CoA synthase and HMG-CoA reductase, which are involved in testosterone synthesis [65]. Our study showed similar findings. The cytosolic HMG-CoA synthase was significantly upregulated in lung tissues of rats exposed to CuO NPs. Zhao et al. postulated that the upregulation of HMSCS with other related proteins, including HMG-CoA reductase and fatty acid synthase, might play a role in attenuating nanoparticle toxicity and facilitating the survival of cells via binding and further reaction following nanoparticle treatment [64]. However, further studies are needed to evaluate the toxicological mechanisms associated with HMSCS.

FIGNL1 is an important member of the ATPase associated with diverse cellular activities and is primarily localized in the nucleus. FIGNL1 has been implicated in numerous biological processes [66,67,68,69,70,71]. FIGNL1 plays an important role in regulating animal developmental morphogenesis and participates in hydrolase, ATPase, microtubule-severing activities, regulation of double-strand break repair by homologous recombination, and promoting non-small cell lung cancer cell proliferation [66,67,68,69,70,71]. In the present study, the FIGNL1 expression was significantly increased in lung tissues of rats exposed to CuO NPs when compared with control groups. It has been reported that CuO NPs induced significant DNA damage and apoptosis in human bronchial epithelial cells and lung tissues [22,46,72,73]. Upregulation of FIGNL1 might protect lung tissues against genotoxicity induced by CuO NPs. Overexpression of FIGNL1 significantly promoted cell proliferation, including decreased arrest at the G1 phase of the cell cycle and apoptosis and increased ability for fission and migration [69].

In conclusion, CuO NPs have emerged as an important class of nanomaterials in a wide range of medicinal, environmental, and industrial applications. In the present study, pulmonary toxicities, including cytotoxicity and immunotoxicity, were evaluated in BALF and proteomics, and oxidative stress studies were performed in lung tissues of rats exposed to CuO NPs. Cytotoxicity and immunotoxicity revealed significant increases in CuO NPs-exposed groups. Proteomic study results led to identification of six proteins, which were up-and downregulated in CuO NPs exposed groups. Taken together, the present study provides additional insight into pulmonary cytotoxicity and immunotoxicity as well as oxidative stress in rats exposed to CuO NPs. Proteomic analysis elucidated the toxicity mechanisms of CuO NPs.

## 4. Materials and Methods

### 4.1. Chemicals

Urea, thiourea, CHAPS, DTT, acrylamide, NN’-methylene-bisacrylamide, iodoacetamide, and sodium thiosulfate were purchased from Sigma Chemicals (St Louis, MO, USA). Protease inhibitor cocktail was purchased from Roche (Mannheim, Germany). HPLC grade solvents, including acetonitrile, acetic acid, and methanol, were purchased from Merck (Merck Co. Darmstadt, Germany). All other chemicals used in this study were the highest grade commercially available.

### 4.2. Characterization and Preparation of CuO NPs

CuO NPs (#544868) were purchased from Sigma-Aldrich (Sigma-Aldrich, St Louis, MO, USA). The CuO NPs were crystalline with a particle size less than 50 nm. The surface area of CuO NPs was 29 m^2^/g. The crystallinity of CuO NPs was investigated via X-ray diffraction using transmission electron microscopy (TEM), HT-7800 (Hitachi, Hitachinaka, Japan) at an accelerating voltage of 120 kV. The particle size and morphology were investigated using a scanning electron microscope (SEM), Quanta FEG 250 (LMA, Zaragoza, Spain) at an acceleration voltage of 15 kV.

### 4.3. Intratracheal Instillation of CuO NPs and Sample Preparation

Specific pathogen-free 7-week-old male Sprague–Dawley rats were purchased from Orient Bio Inc (Seongnam, Korea) and housed under standard animal laboratory conditions at 21 ± 3 °C, 50 ± 20% relative humidity, and a 12 h light/dark cycle. Animals were acclimatized to the facility for 1 week prior to the study. Rats were divided into 3 groups (n = 5 individuals per group) with a vehicle control group (treated with sterile saline). Animals were exposed to isoflurane anesthesia, followed by intratracheal instillation of CuO NPs: 0.15 mg/kg bw (low exposure group) and 1.5 mg/kg bw (high exposure group). Rats were sacrificed 24 hours after the instillation, and the lungs were excised. BALF from rats was obtained via repeated whole-lung lavage. The lungs were lavaged five times with 5 mL each of sterile saline (recovery of BALF > 90%). The BAL fluid was centrifuged at 1500 rpm for 10 min. The volume recovered from the cell-free supernatant of the first lavage fluid was separated from the other samples for further analyses. All animal experiments were accomplished in accordance with the guidelines issued by the Institutional Animal Care and Use Committee of the National Institute of Environmental Research (Approval #NIER-13-2).

### 4.4. Determination of Total Cell Count and Percentages of PMNs, Lymphocytes, Eosinophils, and Macrophages in BALF

Recovered cells from all lavages were resuspended in phosphate-buffered saline. The total number of cells in the BALF were counted with a Vi-Cell ® XR analyzer (Beckman Coulter, Brea, CA, USA), and cell differential tests were performed using cytospin preparations (Shandon, Pittsburgh, PA, USA) stained with Diff-Quick staining solution (Fisher Scientific, Swedesboro, NJ, USA). BALF cells (500/rat) were differentially counted as PMNs, lymphocytes, eosinophils, and macrophages via light microscopy (Olympus Co, Tokyo, Japan).

### 4.5. Determination of LDH, Total Protein Levels, and Inflammatory Cytokines in BALF

The levels of LDH and total protein in BALF were measured to evaluate cytotoxicity and vascular permeability, respectively. LDH levels were measured using a QuantiChrom Lactate Dehydrogenase Kit (BioAssay Systems, Hayward, CA, USA). The total protein (TP) content in the BALF supernatant was quantified using a BCA protein assay kit (iNtRON Biotechnology, Seoul, Korea). The levels of TNF-α and MIP-2 in the BALF were estimated using commercially available enzyme-linked immunosorbent assay (ELISA) kits (R&D Systems, Minneapolis, MN, USA and MIP-2: Invitrogen Life Technologies, Carlsbad, CA, USA), based on the manufacturer’s instructions.

### 4.6. Histopathological Analysis

The right lung tissues were fixed in 10% formalin and then embedded in paraffin. Tissue slides of 3-μm-thick sections were stained with hematoxylin and eosin and then visualized via light microscopy (Olympus Co., Tokyo, Japan).

### 4.7. Sample Preparation and 2-DE Large Gel Polyacrylamide Gel Electrophoresis (PAGE)

Sample preparation and 2-DE large gel PAGE were performed as described previously with minor modifications [74]. Lung tissues finely ground under liquid nitrogen were homogenized via sonication in a lysis buffer containing protease inhibitors, and the insoluble cellular debris was removed by centrifugation. Aliquots of protein samples were prepared as described in the 2D Clean-up Kit (GE Healthcare Life Science, MA, USA), and the pellet obtained in the final step was solubilized in the sample buffer containing 7 M urea, 2 M thiourea, 40 mM Tris (0.5 M, pH 8.5), 4% CHAPS, 65 mM DTT, and 1% IPG buffer. Protein concentrations were measured via Bradford assay (Bio-Rad, CA, USA). Urine was centrifuged at 2500 g for 30 min at 4˚C. The supernatant was collected and concentrated using a Microcon 3,000 molecular weight cutoff filter (Millipore, Bedford, MA, USA) following three washes with 300 μl of 50 mM Tris buffer. In the first dimension of the 2-DE, the proteins were separated according to their isoelectric points. The protein sample (250–300 µg) was mixed with a rehydration buffer (GE Healthcare Life Science, MA, USA) to a total volume of 150 µL per sample. Isoelectric focusing (IEF) was carried out at commercially available immobilized pH gradients (pH 3–11 nonlinear, 3–5.6 nonlinear, 4–7 linear, 6–9 linear, 24 cm), using the IPG Phor (Amersham Biotech, Amersham, UK) apparatus. Next, the IPG gel strips were equilibrated twice for 15 min under gentle shaking at room temperature, first in a solution (equilibration buffer: 50 mM Tris-HCl, pH 8.8, 6 M urea, 30% glycerol, 1% w/v sodium dodecyl sulfate (SDS)) containing 1% dithiothreitol (DTT), followed by an equilibration buffer containing 2.5% iodoacetamide. In the second dimension of SDS-PAGE, the proteins were resolved on 12.5% polyacrylamide gels (size 35 × 45 cm) solely on the basis of their molecular masses using a large-gel separation system. The IPG strips were embedded in 0.5% w/v melted agarose prior to running on the SDS-PAGE slabs. The agarose contained 0.001% w/v BPB as a tracking dye. The running conditions were 1 w/gel for 30 min and 20 w/gel for 14–16 h until the BPB reached the end of the gel.

### 4.8. Visualization and Image Analysis 

Visualization and image analysis were performed as described previously [74]. Following SDS-PAGE, the proteins were visualized via silver staining according to the manufacturer’s instructions (GE Healthcare Life Science, MA, USA). The silver-stained gels were scanned using a 3600 × 4900 dpi instrument (Epson Expression 10000XL, Epson, Japan), and the JPEG files were transformed into a TIFF format with linear grayscale values. The 2DE images were analyzed via Progenesis Samespot image analysis software (Nonlinear Dynamics, Newcastle upon Tyne, UK) according to the manufacturer’s protocol. Intensity levels were normalized between gels as a proportion of the total protein intensity detected on the entire gel.

### 4.9. Identification of Proteins via UPLC-ESI-MS^E^

Protein identification and data analysis were performed as described previously [74]. Briefly, differentially expressed spots were excised from the gels, washed with 10 mM NH_4_HCO_3_ and 50% CH_3_CN (Sigma, St Louis, MO, USA), and digested in a buffer containing 50 mM NH_4_HCO_3_, 5 mM CaCl_2_, and 12.5 ng/ml Trypsin Gold (Promega, Madison, WI, USA) at 37 °C for 12–16 h. The digested peptides were recovered by extraction twice with 50 mM NH_4_HCO_3_ and 100% CH_3_CN. The resulting peptide extracts were pooled, lyophilized in a vacuum centrifuge, and stored at 4 °C for subsequent protein identification. The protein separation was carried out using a nano-ACQUITY Ultra Performance LC Chromatography™ System (Waters Corporation, Milford, MA, USA) with a nano- ACQUITY UPLC BEH130 C18 75 μm × 250 mm column with a particle size of 1.7 μm (Waters) and enriched on a Symmetry C18 RP (180 μm × 20 mm, particle size 5 μm). Five microliters of tryptic-digested peptides in mobile phase A (water with 0.1% formic acid) were loaded onto the column in each experiment, and a 5–45% mobile phase B (acetonitrile with 0.1% formic acid) was used for over 55 min with a step gradient (flow rate of 280 nL/min), followed by an increase to 90% B in 10 min. The eluting peptides were analyzed in the positive ionization mode using the data independent MS^E^ mode. Tandem mass spectroscopy (MS) of [Glu1]-fibrinopeptide (400 fmol/μL) was used to calibrate the time-of-flight analyzer in the range of 50–1990 m/z. [Glu1]-fibrinopeptide (785.8426 m/z) was used for lock mass correction. The collision energy of the low-energy MS mode and elevated-energy mode was set to 4 eV and the 15–40 eV energy ramping mode, respectively. One cycle of MS and MS^E^ was performed every 3.2 s. In each cycle, the MS spectra were acquired for 1.5 s with a 0.1 s interscan delay (50–1990 m/z), and ions exceeding 50 counts were selected for MS^E^ fragmentation in the collision cell (50–1990 m/z). To generate ion spectra for subsequent database search, ProteinLynx Global Server version 2.4 was used to analyze the liquid chromatography (LC)-MS^E^ raw data files. Proteins were identified based on the search on the UniProt website from the Rattus norvegicus (Rat) database (28,063 entries). For the ProteinLynx Global Server search, low or high collision spectra were analyzed using a hierarchical approach, which required detection of at least three fragment ion matches per peptide, seven fragment ion matches per protein, and two peptide matches per protein with a maximum 4% false positive rate. Cysteine carbamidomethylation (+57 Da) and methionine oxidation (+16 Da) were selected as the fixed and variable modifications, respectively.

### 4.10. Western Blot Assay

Lung tissue samples were prepared using the lysis buffer containing protease inhibitors. The protein sample (5–30 µg) was diluted with a 2X sample loading buffer and denatured by heating for 5 min between 70 and 100˚C. In Western blot analysis, the Bio-Rad Western blot system, including a Bio-Rad Western blot kit (Criterion cells, size: 13 × 8.7 cm, 18 wells) and a Bio-Rad transfer kit (Criterion Blotter, size: 15 × 9.4 cm), was used to show expression patterns of all protein samples obtained from control and case subjects. There were 15 protein samples (5 samples for each of three groups: control, 0.15 mg/kg bw, and 30 mg/kg bw CuO NPs). First of all, the condition of Western blot analysis was performed according to information of antibody catalogs provided by companies. After protein samples were subjected to SDS-PAGE with a running gel kit, proteins in running gels were transferred onto PVDF membranes. PVDF membranes containing target protein bands were cut prior to hybridization with antibodies. PVDF membranes were blocked with 5% skim milk in PBS-T for 1 h and incubated at room temperature for 1 h with primary antibodies specific to catalase (1:500, Santa Cruz), Gpx-1 (1:1000, Abcam), peroxiredoxin-2 (Prx-2) (1:500, Santa Cruz), HSP 10 (1:500, Santa Cruz), HSP 40 (1:1000, Cell signaling), HSP 47 (1:500, Santa Cruz), HSP 70 (1:500, Santa Cruz), HSC 70 (1:500, Santa Cruz), HSP 90 (1:1000, Cell signaling), ANK repeat and PH domain-containing protein 2 (ASAP2) (Santa Cruz, 1:1000), fidgetin-like 1 (FIGNL1) (1:500, Santa Cruz), annexin II (1:500, Santa Cruz), cytosolic HMG-CoA synthase (cHMGCS) (1:500, Santa Cruz), proteasome α1 (1:500, Santa Cruz), and β-actin (Santa Cruz, 1:2000). After washing the membranes with PBS-T three times for 5 min each, they were incubated with HRP-conjugated secondary antibodies [anti-rabbit IgG, anti-goat IgG or anti-mouse IgG (1:2000, Santa Cruz, CA, USA)] for 1 h with rocking and then washed with PBS-T 3 times for 20 min. The membranes were incubated with Pierce™ ECL Western Blotting Substrate (Themo fisher, USA) for 5 min with rocking. Bands were imaged and scanned using a 3600 × 4900 dpi instrument (Epson Expression 10000XL, Epson, Nagano, Japan), followed by analysis using the Image J program.

### 4.11. Statistical Analysis

Statistical analysis was performed with IBM SPSS statistics version 22.0 software (IBM, Inc., Chicago, IL, USA). All data are expressed as the mean ± SD. Between-group differences were analyzed using the non-parametric Mann–Whitney U test. The statistical significance level is indicated in the figures and tables at * *p* < 0.05, ** *p* < 0.01, and *** *p* < 0.001.

## Figures and Tables

**Figure 1 ijms-23-13265-f001:**
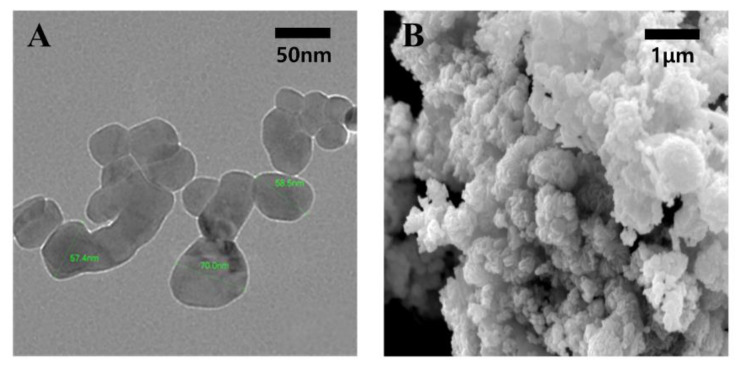
TEM and SEM analyses of CuO NPs: (**A**) TEM analysis at an accelerating voltage of 120 kV (magnification 80,000); (**B**) SEM analysis at an acceleration voltage of 15 kV (magnification, 20,000×).

**Figure 2 ijms-23-13265-f002:**
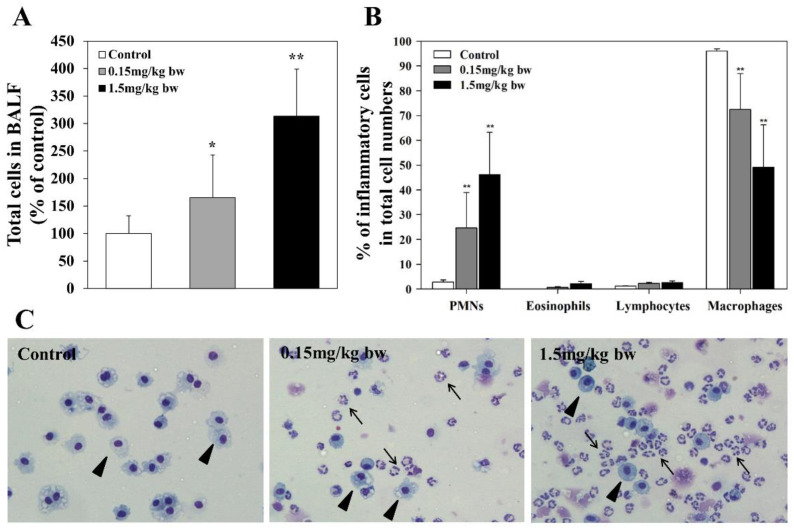
Total cell count in BALF and percentages of inflammatory cells in total cells of rats exposed to CuO NPs: (**A**) total cell count in BALF; the total number of cells in the BALF were counted with a Vi-Cell^®^ XR analyzer (Beckman Coulter, Brea, CA, USA); (**B**) PMNs, eosinophils, lymphocytes, and macrophages in BALF; (**C**) BALF cells via light microscopy. Cell differential tests were performed using cytospin preparations (Shandon, Pittsburgh, PA, USA) stained with Diff-Quick staining solution (Fisher Scientific, Swedesboro, NJ, USA), respectively. BALF cells (500/rat) were differentially counted via light microscopy (Olympus Co, Tokyo, Japan). Arrow: PMN; Arrow head: alveolar macrophage (magnification, 400×).All results are presented as the means ± SD of five experiments. All relative intensity results are expressed as the means ± SD of five experiments; * and ** indicate *p*-values of 0.05 and 0.01, respectively, compared to the control.

**Figure 3 ijms-23-13265-f003:**
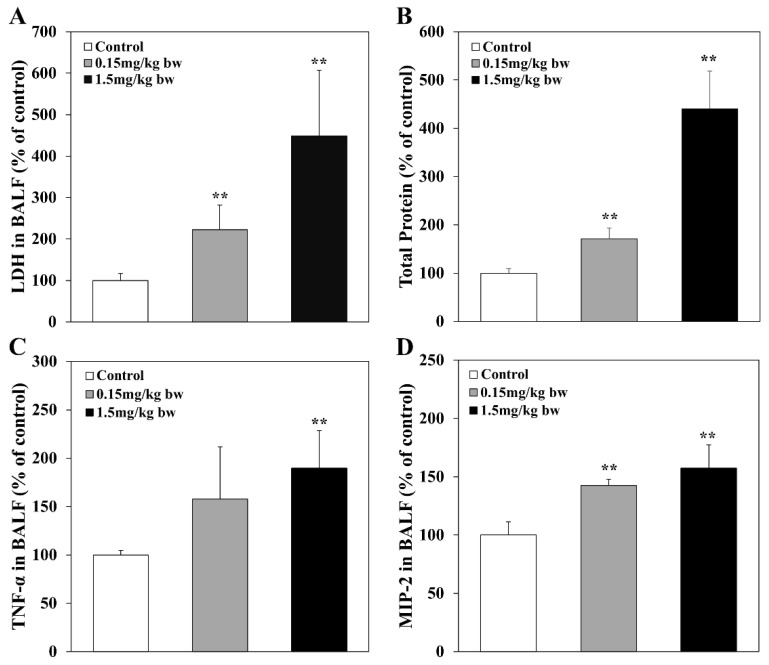
LDH activity, total proteins, and inflammatory cytokines in BALF of rats exposed to CuO NPs: (**A**) LDH in BALF, LDH activities were measured using a QuantiChrom Lactate Dehydrogenase Kit (BioAssay Systems, Hayward, CA, USA); (**B**) total protein in BALF. The total protein in the BALF supernatant was quantified using a BCA protein assay kit (iNtRON Biotechnology, Seoul, Korea). The levels of inflammatory cytokines in BALF were measured using commercially available enzyme-linked immunosorbent assay (ELISA) kits (R&D Systems, Minneapolis, MN, USA and MIP-2: Invitrogen Life Technologies, Carlsbad, CA, USA), based on the manufacturer’s instructions; (**C**) TNF-α; and (**D**) MIP-2. All results are presented as the means ± SD of five experiments; ** indicates *p*-values of 0.01 compared with the control.

**Figure 4 ijms-23-13265-f004:**
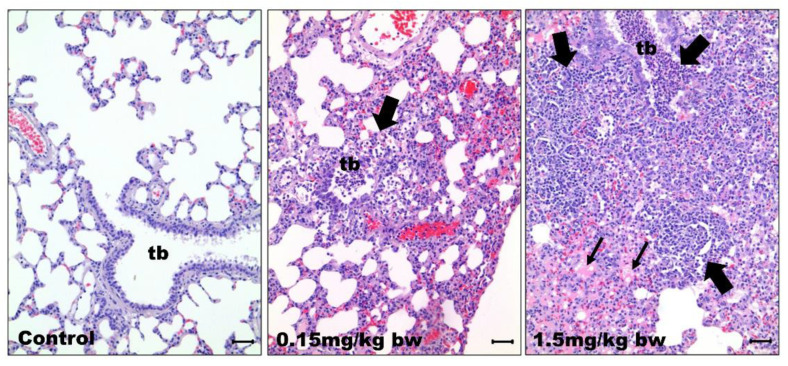
Histopathological observations in lungs of rats exposed to CuO NPs. Lung tissues were fixed in 10% formalin and then embedded in paraffin. Tissue slides of 3-μm-thick sections were stained with hematoxylin and eosin and then visualized via light microscopy: control group (saline as vehicle only); 0.15 mg/kg bw CuO NPs; 1.5 mg/kg bw CuO NPs; tb: terminal bronchiole; thick arrow: accumulation and infiltration of neutrophils and mononuclear cells; thin arrow: alveolar spaces filled with edematous fluid. Bars = 50 µm.

**Figure 5 ijms-23-13265-f005:**
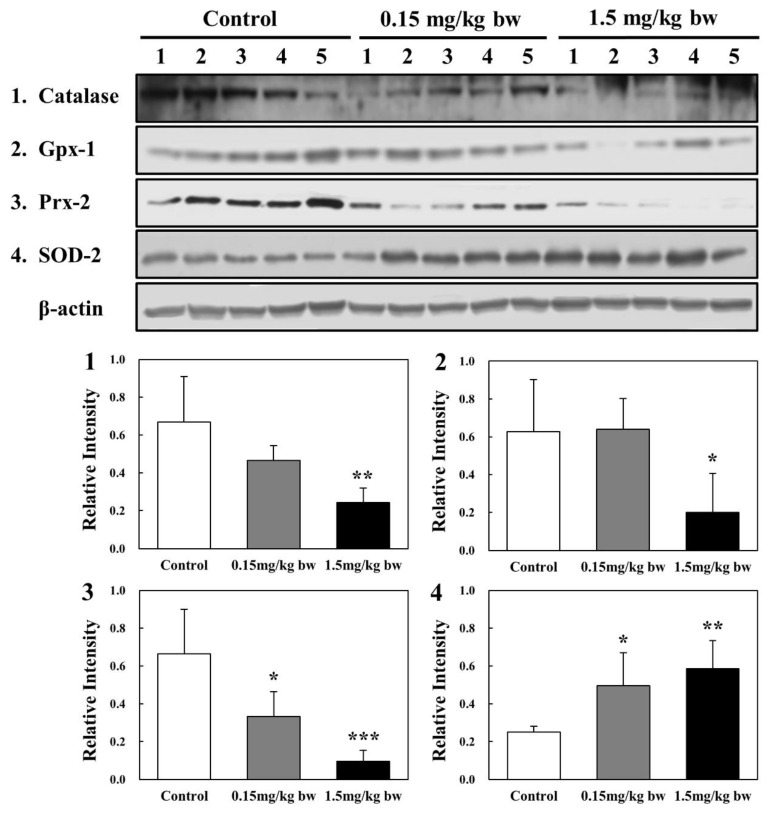
Western blot analysis of antioxidant enzymes in rat lung tissues: (**1**) catalase; (**2**) Gpx-1; (**3**) Prx-2; and (**4**) SOD-2. Amounts represented by the gel bands are expressed as intensity relative to β-actin. All relative intensities are presented as means ± SD of five experiments; *, **, and *** indicate *p*-values of 0.05, 0.01, and 0.001, respectively, compared with the control.

**Figure 6 ijms-23-13265-f006:**
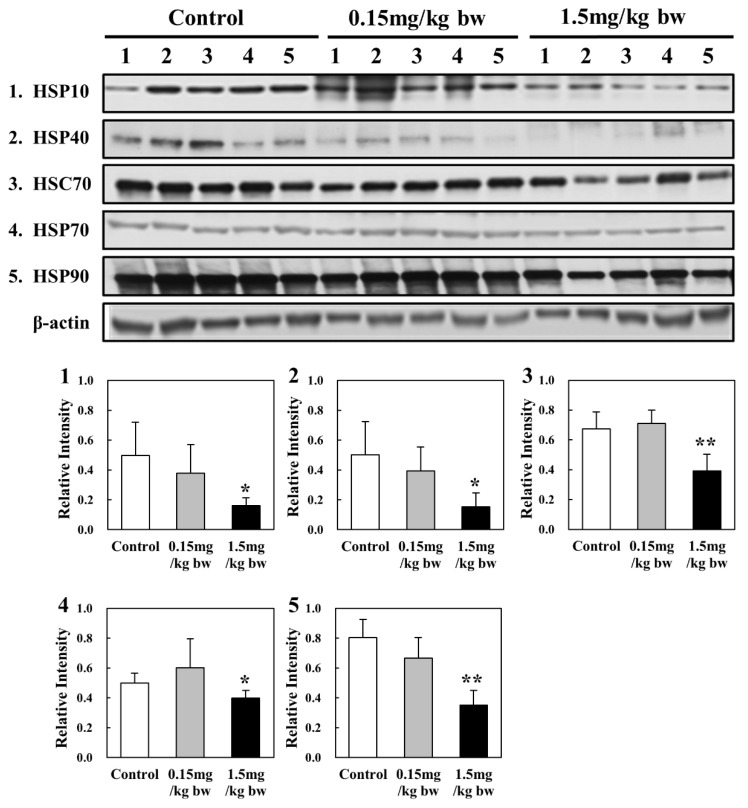
Western blot analysis of HSPs in rat lung tissues: (**1**) HSP 10; (**2**) HSP 40; (**3**) HSC 70; (**4**) HSP 70; and (**5**) HSP 90. Quantities represented by the gel bands are expressed as intensities relative to β-actin. All relative intensities results are presented as the means ± SD of five experiments; * and ** indicate *p*-values of 0.05 and 0.01, respectively, compared with the control.

**Figure 7 ijms-23-13265-f007:**
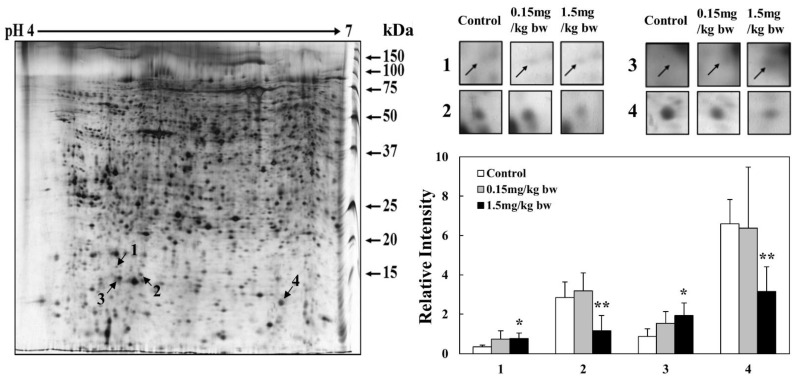
2-DE analysis of lung tissue proteins obtained using 4–7 pI range strips. Gel patterns of tissue proteins obtained using 4–7 pI strips. Protein spot images were analyzed using the Progenesis Samespot software program (Nonlinear Dynamics, Newcastle upon Tyne, UK). Spot volumes were normalized by comparison with the total spot volume. The quantity of each spot is presented as the relative intensity. The images represent the means ± SD of five separate experiments; * and ** indicate *p*-values of 0.05 and 0.01, respectively, compared with the control.

**Figure 8 ijms-23-13265-f008:**
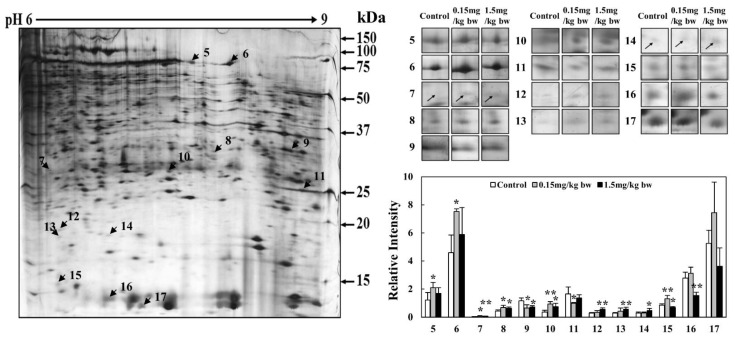
2-DE analysis of kidney tissue proteins obtained using 6–9 pI range strips. Gel pattern of tissue proteins obtained using 6–9 pI strips. Protein spots were analyzed using the Progenesis Samespot software program (Nonlinear Dynamics, Newcastle upon Tyne, UK). Spot volumes were normalized by comparison with the total spot volume. The quantity of each spot is presented as the relative intensity. The images represent the mean ± SD of five separate experiments; * and ** indicate *p*-values of 0.05 and 0.01, respectively, compared with the control.

**Figure 9 ijms-23-13265-f009:**
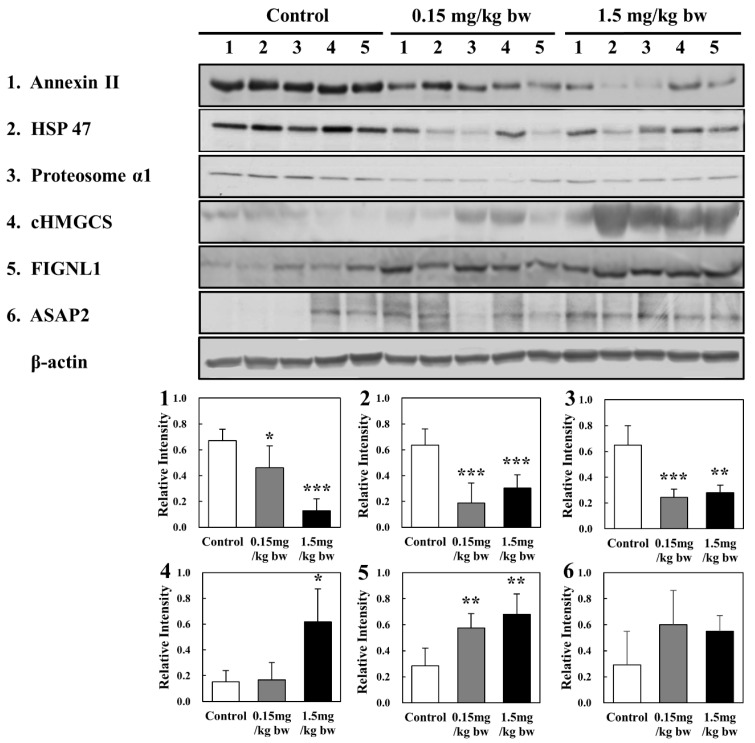
Confirmation of identified proteins by Western blot analysis: (**1**) Annexin II; (**2**) HSP 47; (**3**) 20s proteosome α1; (**4**) HMGCS; (**5**) FIGNL1; and (**6**) ASAP2. Quantities represented by the gel bands are expressed as intensities relative to β-actin. All relative intensities are presented as the means ± SD of five experiments; *, **, and *** indicate *p*-values of 0.05, 0.01, and 0.001, respectively, compared with the control.

**Table 1 ijms-23-13265-t001:** UPLC-ESI-MSE identification of differentially expressed secreted proteins in the lungs of rats exposed to CuO NP.

Spot No.	Accession No.	Protein Name	Fold Change (mg/kg bw)		Matched Peptides	Sequence Coverage (%)
			0.15	1.5		
1	P17425	Hydroxymethylglutaryl CoA synthase cytoplasmic	+2.11	+2.22	3	10.2
2	F7FIH7	Protein LOC100909412	+1.10	−2.43	11	37
3	Q6GX84	Fidgetin like protein 1	+1.77	+2.23	4	7.8
4	M0RBF1	Complement C3	−1.04	−2.05	138	30.1
5	D3ZDK8	Uncharacterized protein	+1.71	+1.38	4	23.5
6	P02091	Hemoglobin subunit beta 1	+1.63	+1.27	10	47.6
7	F1LZT6	Protein Asap2 Fragment	+2.50	+1.83	6	6.6
8	F1LWD5	Protein 1700056E22Rik	+1.58	+1.42	3	9.4
9	M3ZCQ3	Protein LOC100910765	−1.78	−1.60	14	79.3
10	Q3B8N7	TSC22 domain family protein 4	+2.50	+2.10	6	13.2
11	O35817	A kinase anchor protein 14	−1.70	−1.21	7	4.8
12	D3Z8I7	Protein Gstt3	+1.13	+1.88	8	19.5
13	P01946	Hemoglobin subunit alpha 1 2	+1.57	+2.14	19	48.6
14	F1LQ93	Collagen alpha 1 IX chain	+1.14	+1.86	5	11.2
15	P29457	Serpin H1	+1.13	−1.21	9	22.5
16	Q07936	Annexin A2	+1.13	−1.79	19	45.1
17	P18420	Proteasome subunit alpha type 1	+1.41	−1.46	12	31.9

## Data Availability

The data presented in this study will be made available on request to the corresponding author.

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
