# Peer review of "Pulmonary Toxicity and Proteomic Analysis in Bronchoalveolar Lavage Fluids and Lungs of Rats Exposed to Copper Oxide Nanoparticles"

_ijms, 2022, doi:10.3390/ijms232113265_

Round 1

Reviewer 1 Report

Comments to the Author:

The authors investigated the pulmonary cytotoxicity and immunotoxicity of CuO nanoparticles. There is great work being done and a lot of data shown. Authors need to add additional discussions and interpretation of results.  Some clarification is also needed for the SI data and introductions section. There are certain aspects, which should be improved.

·       • Authors should choose and use one abbreviation for CuO nanoparticles (CuO NP or CuO NPs).

·       • line 47: Incorrect definition of organic nanoparticles. Authors should more accurately apply the definition and categorization used in nanotechnology.

·       • Did authors compare their results for CuO NPs with data copper ions?

·       • The following statement should be carefully checked: “Their impact on living cells and tissues is attributed to their size and shape rather than the chemical behavior of specific metal ions used [2].” Tha statement taken from review which gives reference on another review. Authors should give reference on the original article with approved data.

·       • Clarification and short description for Table S1 (Supplementary protein identification data) is desirable.

·       • line 81: References are missing.

·       • line 255-284: There are results descriptions, some discussions are desirable.

Author Response

Reviewer # 1 comments

The authors investigated the pulmonary cytotoxicity and immunotoxicity of CuO nanoparticles. There is great work being done and a lot of data shown. Authors need to add additional discussions and interpretation of results.  Some clarification is also needed for the SI data and introductions section. There are certain aspects, which should be improved.

  1. “Authors should choose and use one abbreviation for CuO nanoparticles (CuO NP or CuO NPs)”

  • We apologize for the confusion. It should be “CuO NPs” for CuO nanoparticles. We have changed “CuO NP” to “CuO NPs” throughout the manuscript.

  1. “line 47: Incorrect definition of organic nanoparticles. Authors should more accurately apply the definition and categorization used in nanotechnology.”

  • We thank the reviewer for pointing this out and we agree with the reviewer. Therefore, we have deleted previous information and rewritten the classification of nanoparticles. We have also focused on inorganic nanoparticles with suitable reference (Line 45-49).

  1. “Did authors compare their results for CuO NPs with data copper ions?”

  • We did not compare our results for CuO NPs with copper ions data. Previously, a few comparative toxicological studies on Cu ions and CuO NPs have been reported. One in vitro study showed that Cu in the form of CuO NPs had eight times higher cytotoxicity to CuCl2 A549 lung cells than Cu ions [Karlsson et al., Chem. Res. Toxicol. 2008]. On the other hand, CuO NPs and CuCl2 showed similar levels of LDH and total proteins of BALF in rats exposed by intratracheal instillation [Jeong et al., Nanotoxicology 2016]. If the reviewer wants us to show this information in the manuscript, we will add the previous comparative studies on Cu ions and CuO NPs in the Discussion section.

  1. “The following statement should be carefully checked: “Their impact on living cells and tissues is attributed to their size and shape rather than the chemical behavior of specific metal ions used [2].” The statement taken from review which gives reference on another review. Authors should give reference on the original article with approved data.”

  • We thank the reviewer for pointing this out and we agree with the reviewer. Therefore, we have revised it and provided the original reference (Kapoor et al., 2018 J Environ Sci Health A Tox Hazard Subst Environ Eng.) (Line 50-51).

  1. “Clarification and short description for Table S1 (Supplementary protein identification data) is desirable”

  • We thank the reviewer for pointing this out and we agree with the reviewer. Therefore, we have included a new supplementary table for protein identification data. The new supplementary table shows additional data including root mean square errors, protein ID, theoretical peptides, molecular weight, and pH that were not provided in Table 1 (Supplementary Table S1).

  1. “line 81: References are missing”

  • We have searched proteomic studies on CuO NPs in rat lungs. However, there were no proteomics data of CuO NPs. Several proteomic studies only reported data using BEAS-2B human lung cells and other organisms including mussel Mytilus galloprovincialis, Mytilus edulis and Cucumis sativus seeds. We have added these studies as references (Line 79).

  1. “line 255-284: There are results descriptions, some discussions are desirable”

  • We thank the reviewer for pointing this out and we agree with the reviewer. Therefore, we have deleted such result description and rewritten the discussion part (Line 256-272).

We thank the reviewer for the helpful comments that have helped to improve the quality of this paper. We hope that you will see satisfied with our response and with the changes made to the revised manuscript.

Very truly yours

Donggeun Sul, Ph.D.

Professor

Graduate School of Medicine,

Environmental Toxico-Genomic and Proteomic Center,

Korea University,

126-1, Anam-Dong 5-Ka, Sungbuk-Ku,

Seoul, 136-705 Republic of Korea

Tel: 82-2-922-6174,

Fax: 82-2-927-7220,

Reviewer 2 Report

Generally, the results of the study are interesting and the methods used are appropriate.  Cytokines are generally pro- or anti-inflammatory, and the balance between these determines the outcome of an inflammatory response. IL-1β, IL-8, and IFN-γ for example, are pro-inflammatory cytokines involved in early responses and the amplification of inflammatory reactions. Anti-inflammatory cytokines like IL-4, IL-10, and IL-13 limit inflammatory responses. Cytokines do not operate in isolation but rather in a network of activation and suppression. Assessing inflammatory responses via the measurement of individual cytokines can often fail to reflect this complexity. Thus, it would have been interesting to evaluate the cytokines mentioned above. In addition, in the manuscript there are cropped blots displayed (Figures 5, 6 and 9), we suggest that full-length blots to be included in a Supplementary Information file.

Author Response

Reviewer # 2 comments

  • “Generally, the results of the study are interesting and the methods used are appropriate. Cytokines are generally pro- or anti-inflammatory, and the balance between these determines the outcome of an inflammatory response. IL-1β, IL-8, and IFN-γ for example, are pro-inflammatory cytokines involved in early responses and the amplification of inflammatory reactions. Anti-inflammatory cytokines like IL-4, IL-10, and IL-13 limit inflammatory responses. Cytokines do not operate in isolation but rather in a network of activation and suppression. Assessing inflammatory responses via the measurement of individual cytokines can often fail to reflect this complexity. Thus, it would have been interesting to evaluate the cytokines mentioned above.”

  • We thank the reviewer for pointing this out and we agree with the reviewer. In inflammatory responses to metal and metal oxide nanoparticles, detailed investigations of pro- and anti-inflammatory cytokines are needed to elucidate the network of activation and suppression for inflammatory responses. In a previous study, Cho et al. have investigated the profile of inflammatory mediators for neutrophilic (TNF-α, IL-1β, and MIP-2), eosinophilic (eotaxin and IL-13), and lympho-cytic (IFN-γ) inflammation in BALF of rats exposed to various metal oxide nanoparticles including CeO2 NPs, NiO NPs, ZnO NPs, and CuO NPs (Cho et al., Environ Health Perspect. 2010). In that study, they demonstrated that each of these metal oxide nanoparticles induced a unique inflammatory footprint both acutely (24 hr) and chronically (4 weeks). However, their study did not elucidate the network of activation or suppression of inflammatory responses. In the present study, we only evaluated levels of TNF-α and MIP-2 for immunotoxicity of CuO NPs in BALF of rats. Our study has a limitation for elucidating inflammatory responses in rat lungs exposed to CuO NPs. As mentioned by reviewer#2, we need a further study to investigate inflammatory responses by determining pro- and anti-inflammatory cytokines (IL-1β, IL-8, IL-4, IL-10, IL-13 and IFN-γ) and their balance. We have added this as a limitation of our study in the Discussion section of the revised manuscript (Line 285-288).

  • “In addition, in the manuscript there are cropped blots displayed (Figures 5, 6 and 9), we suggest that full-length blots to be included in a Supplementary Information file”

  • In Western blot analysis, we used the Bio-Rad Western blot system including a Bio-Rad western blot kit (Criterion cells, size: 13 X 8.7 cm, 18 wells) and a Bio-Rad transfer kit (Criterion Blotter, size: 15 X 9.4 cm) because we wanted to show expression patterns of all protein samples obtained from control and case subjects. There were 15 protein samples (5 samples for each of three groups: Control, 0.15 mg/kg BW, and 30 mg/kg BW CuO NPs). First of all, the condition of Western blot analysis was based on the information of antibody catalogs provided by companies. After protein samples were subjected to SDS-PAGE with a running gel kit, proteins on gels were transferred onto PVDF membranes. PVDF membranes containing target protein bands were cut prior to hybridization with antibodies. Unfortunately, we could only add western blot data in this study. We have added the methodological information for Western blot analysis in the Materials and Methods section (Line 542-551).

We thank the reviewer for the helpful comments that have helped to improve the quality of this paper. We hope that you will see satisfied with our response and with the changes made to the revised manuscript.

Very truly yours

Donggeun Sul, Ph.D.

Professor

Graduate School of Medicine,

Environmental Toxico-Genomic and Proteomic Center,

Korea University,

126-1, Anam-Dong 5-Ka, Sungbuk-Ku,

Seoul, 136-705 Republic of Korea

Tel: 82-2-920-6614,

Round 2

Reviewer 1 Report

The authors improved the manuscript and  I found that they have made good corrections and modifications in the work as per my suggestions. Unfortunately the work still have the following concerns:
line 55-57: Author should not use the inaccurate data in the article. Problem is not the reference. The statement is initially incorrect. The information requires the correct proofs and relevant references.

Author Response

Reviewer # 1 comments

The authors improved the manuscript and I found that they have made good corrections and modifications in the work as per my suggestions. Unfortunately the work still have the following concerns:

line 55-57: Author should not use the inaccurate data in the article. Problem is not the reference. The statement is initially incorrect. The information requires the correct proofs and relevant references.

  1. “line 55-57: Author should not use the inaccurate data in the article. Problem is not the reference. The statement is initially incorrect. The information requires the correct proofs and relevant references.”

  • We apologize for giving the incorrect information. In this study, we had tried to provide more general information of metal and metal oxide nanoparticles that was described in the previous review paper (Chaudhary et al., Curr. Pharm. Des. 2019). Originally, this information was referred from another review paper (Kapoor et al., J. Environ. Sci. Health A Tox. Hazard. Subst. Environ. Eng. 2018). We have searched the information of metal and metal oxide nanoparticles in the original paper. However, we did not find it. We agree with the reviewer#1. We deleted this part from Introduction (Line 55-57).

We thank the reviewer for the helpful comments that have helped to improve the quality of this paper. We hope that you will see satisfied with our response and with the changes made to the revised manuscript.